# Modification of Collagen/Gelatin/Hydroxyethyl Cellulose-Based Materials by Addition of Herbal Extract-Loaded Microspheres Made from Gellan Gum and Xanthan Gum

**DOI:** 10.3390/ma13163507

**Published:** 2020-08-08

**Authors:** Justyna Kozlowska, Weronika Prus-Walendziak, Natalia Stachowiak, Anna Bajek, Lukasz Kazmierski, Bartosz Tylkowski

**Affiliations:** 1Faculty of Chemistry, Nicolaus Copernicus University in Torun, Gagarina 7, 87-100 Torun, Poland; 279502@stud.umk.pl (W.P.-W.); nat.sta@doktorant.umk.pl (N.S.); 2The Ludwik Rydygier Collegium Medicum, Nicolaus Copernicus University, Karlowicza 24, 85-092 Bydgoszcz, Poland; a.kaznica@cm.umk.pl (A.B.); lukasz.kazmierski@cm.umk.pl (L.K.); 3Eurecat, Centre Tecnològic de Catalunya, C/Marcellí Domingo, 43007 Tarragona, Spain; bartosz.tylkowski@eurecat.org

**Keywords:** encapsulation, microspheres, *Calendula officinalis* flower extract, release profile, structure, stability, cytotoxicity, mechanical properties, swelling properties

## Abstract

Because consumers are nowadays focused on their health and appearance, natural ingredients and their novel delivery systems are one of the most developing fields of pharmacy, medicine, and cosmetics. The main goal of this study was to design, prepare, and characterize composite materials obtained by incorporation of microspheres into the porous polymer materials consisting of collagen, gelatin, and hydroxyethyl cellulose. Microspheres, based on gellan gum and xanthan gum with encapsulated *Calendula officinalis* flower extract, were produced by two methods: extrusion and emulsification. The release profile of the extract from both types of microspheres was compared. Then, obtained microparticles were incorporated into polymeric materials with a porous structure. This modification had an influence on porosity, density, swelling properties, mechanical properties, and stability of materials. Besides, in vitro tests were performed using mouse fibroblasts. Cell viability was assessed with the MTT (3-(4,5-dimethylthiazol-2-yl)-2,5-diphenyltetrazolium bromide) assay. The obtained materials, especially with microspheres prepared by emulsion method, can be potentially helpful when designing cosmetic forms because they were made from safely for skin ingredients used in this industry and the herbal extract was successfully encapsulated into microparticles.

## 1. Introduction

The encapsulation method was developed in the 1950s. The first microencapsulation process was invented in 1953 (B. K. Green and L. Schleicher) in the National Cash Register Company laboratory. This patent concerned the encapsulation of leuko dyes for self-copy papers. Over the past few decades, the interest in the technique of encapsulation has extremely increased. Microparticles are spherical particles with a diameter in the range of 1 µm to 1000 µm formed from synthetic or naturally occurring polymers [1,2]. Extracellular polymeric substances such as gellan gum [3,4], xanthan gum [5], alginate [6], hyaluronic acid [7], chitosan [8], and cellulose [9] or their mixtures have attracted the scientist’s attention. Xanthan gum is a microbial polysaccharide produced by the bacterium *Xanthomonas campestris* through the fermentation process [10,11]. It has a variety of applications (including in the food, cosmetic, pharmaceutical, medical, and petroleum industries) due to its unique rheological properties stabilizing water-based systems [11,12]. Xanthan gum is resistant to enzymatic degradation and its viscosity is stable over a wide range of pH and temperature [13]. Another polysaccharide secreted by the bacterium (*Sphingomonas elodea*) is gellan gum. It is a linear, anionic polymer used as a microbiological gelling agent, as well as a food additive with strong stabilizing, thickening, and gelling properties [14,15]. Apart from this, gellan gum has been utilized in biomedicine and material science taking the advantage of its peculiar gelling and rheological properties, as well as biocompatibility [16,17,18]. 

The microparticles can be divided into microspheres and microcapsules due to the construction and the method of placing the active substance. Microspheres are particles that are made of an irregular polymer coating and active substance which is incorporated into that polymer matrix [19]. They have a more favorable structure in terms of possible damage because, in case of breakage of the coating, the active substance will not be completely released. Microcapsules, on the other hand, are particles that are characterized by a continuous core surrounded by a polymer coating. In contrast to microspheres, active substances are not distributed but enclosed in microcapsules [19,20]. In the case of damage to the shell, the entire substance is released from the matrix. The release of the active substance from the microparticles occurs as a result of the slow enzymatic degradation of the polymer coating.

Microparticles have an ability to encapsulate various kinds of substances such as solids, liquids, or gaseous materials [20,21]. Therefore, they have found application in a wide range of science branches, especially medicine, pharmacy, cosmetics, and food industry [22,23,24,25,26,27]. Encapsulation is a very useful tool to enhance the delivery and to facilitate sustained and strictly controlled release of encapsulated substances to the proper site with the required rate or optimal kinetics for a long period [1,28,29,30,31]. It also enables to protect these substances against the damaging effects of external factors. 

Incorporation of microspheres into porous polymer matrices enables to obtain composite materials which combine the advantages of microspheres and the polymeric matrix. Modern materials used in cosmetics, medicine, and pharmacy should be biodegradable, non-toxic, and biocompatible. Therefore, more and more attention is focused on biopolymers such as collagen, gelatin, and hydroxyethyl cellulose, which may be used to design materials with appropriate properties [32,33,34]. Collagen is the main structural protein responsible for the structural integrity of the connective tissue of many multicellular organisms. This protein, as a building material, shows a broad versatility - it provides mechanical stability, toughness, tensile strength, and elasticity. Gelatin is a denatured, biodegradable protein obtained by processing of collagen extracted from animal tissue such as skin, muscle, and bone. This biopolymer exhibits excellent gelling properties [35,36]. Hydroxyethyl cellulose is a cellulose derivative and is used as a thickener and protective colloid [37]. The combination of suitable polymers allows the improvement in functional and application properties of materials.

The aim of our study was to obtain and characterize composite materials for cosmetic by incorporation microspheres containing *Calendula officinalis* flower extract into the biopolymeric matrices with porous structure. The microspheres were obtained from gellan gum or gellan gum and xanthan gum mixtures with the use of extrusion or emulsion method. Subsequently, prepared microspheres were incorporated into matrices consisting of collagen, gelatin and hydroxyethyl cellulose with different ratios. This type of matrix was described in our previous work with promising results [38], so we decided to investigate how it will work after modifying with another type of microparticles. The fabricated materials were characterized by porosity, density, swelling properties, mechanical properties, and in vitro degradation. The cytotoxicity of the tested materials was tested using MTT assay.

## 2. Materials and Methods 

### 2.1. Materials

Collagen type I (COL) was obtained in our laboratory from scales of freshwater fish *Esox lucius* (Dejguny Lake, Poland) [39]. Gelatin (GEL) from porcine skin, hydroxyethyl cellulose (HEC), gellan gum (GG), xanthan gum (XG), 1-ethyl-3(3-dimethylaminopropyl) carbodiimide (EDC), N-hydroxysuccinimide (NHS), Tween 80, Folin–Ciocalteu reagent and gallic acid were acquired from Sigma-Aldrich (Poznan, Poland). The hydroglycolic *Calendula officinalis* flower extract was obtained from Provital S.A. (Barcelona, Spain). All the other reagents were obtained from Avantor Performance Materials Poland S.A. (Gliwice, Poland). All used chemicals were of analytical grade.

### 2.2. Microencapsulation Procedures

The microspheres were produced from gellan gum and xanthan gum with incorporated *Calendula officinalis* flower extract by two methods: extrusion and emulsification. 

First, gellan gum solution (1.5%) in 0.1% *Calendula officinalis* flower extract and mixtures of gellan gum (1.5%) and xanthan gum (0.25%) in 0.1% *Calendula officinalis* flower extract were made. Then, they were poured into a syringe with a needle (diameter of 0.6 mm) to 0.5 M CaCl_2_. The microspheres were obtained as a result of the forced flow of the gellan gum and xanthan gum sol from a narrow needle and their gelation in a bath with calcium chloride solution. 

In order to obtain the microspheres by emulsion method, 60 mL of gellan gum solution (1.5%) and mixtures of gellan gum (1.5%) and xanthan gum (0.25%) in 0.1% *Calendula officinalis* flower extract were prepared. After that, 240 mL of paraffin oil and 48 µL of Tween 80 were added to the obtained mixtures by stirring on a mechanical stirrer to form a homogeneous emulsion. Subsequently, 0.5 M CaCl_2_ was added until the resulting emulsion fractured to cross-link the microspheres. 

### 2.3. Matrices Preparation

The polymer matrices were produced from collagen, gelatin and hydroxyethyl cellulose using the method detailed in our previous paper with some modifications [38]. Firstly, collagen and gelatin were dissolved in deionized water, to prepare 0.5% (*w*/*v*) of collagen suspension and 0.5% (*w*/*v*) of gelatin type A solution. After that, mixtures were prepared by mixing suitable volumes of collagen and gelatin solutions with the addition of hydroxyethyl cellulose and the final weight ratio was 25:25:50. In the next step, microspheres were added to these mixtures and the final amount of microspheres was 1.5% and 3% (*w*/*v*) (Table 1). The mixtures were frozen (−20 °C) and lyophilized (−55 °C, 5 Pa, 24 h) using an ALPHA 1–2 LD plus lyophilizator (Martin Christ, Osterode am Harz, Germany). Then, the samples were crosslinked using the crosslinking agents. For this purpose, the matrices were immersed in the mixture containing 96% ethyl alcohol, 50 mM EDC, and 25 mM NHS at room temperature for 4 h. After this time, the mixture was removed and the samples were put into 0.1 M Na_2_HPO_4_ solution for 2 h (changing the solution twice). Then, the matrices were washed with distilled water four times within 30 min. In the end, the crosslinked samples were frozen (−20 °C) and lyophilized (5 Pa, −55 °C, 48 h). The matrices were fabricated with the addition of all the types of the obtained microparticles. The matrix without microspheres was a control sample named COL/GEL/HEC. 

### 2.4. Characterization of Materials

#### 2.4.1. Structure and Morphology of Microspheres and Matrices

The microspheres physical size and appearance were observed by the optical microscope Motic SMZ-171 BLED (Hong Kong, China) in magnification 10×. Diameters of swollen microspheres obtained by extrusion and emulsion methods and the diameters of swollen microspheres produced by the emulsion method were measured by the optical microscope.

Scanning electron microscopy (SEM) imaging was performed using the Quanta 3D FEG scanning electron microscope produced by Quorum Technologies (Lewes, UK) to analyze the structure of obtained three-dimensional materials. Prior to the analysis, the surface of the samples was sprayed with a thin layer of gold and palladium. The diameters of dry microspheres prepared by the emulsion method were measured by SEM after incorporation into polymer matrices and the lyophilization process. 

#### 2.4.2. Loading Capacity of Microspheres

The loading capacity of microspheres was determined by quantifying the polyphenolic compounds contained in the pot marigold extract-loaded microspheres using the Folin–Ciocalteu test [40]. The microspheres were weighed and placed in 2 mL of 1 M NaOH for 1 h. Then, the resulting suspension was centrifuged (10,000 rpm, 5 min) and the supernatant solution was collected. Samples with the extract (20 µL) were mixed with 1.58 mL distilled water and 100 µL Folin–Ciocalteu reagent. After 4 min, 300 µL of saturated Na_2_CO_3_ solution was added. The prepared mixtures were incubated (40 °C, 30 min) until a characteristic blue color was obtained. The absorbance was measured at 725 nm using a UV-Vis spectrophotometer (UV-1800, Shimadzu, Kyoto, Japan). The data of polyphenol content was expressed based on gallic acid using the standard curve equation. The results presented are the average of measurements made for three samples of each type of microspheres.

#### 2.4.3. In Vitro Release

The microspheres were weighed (in triplicate) and placed in 24-well polystyrene plates. Then, 2 mL of acetate buffer (pH = 5.4) was added to each sample. The plates were incubated at 37 °C. The solution was collected after 1, 2, 3, and 4 h—each time adding to the microspheres acetate buffer stored at 37 °C. The obtained samples were frozen (−20 °C) and after collecting all the samples, the content of phenolic compounds was determined using the Folin–Ciocalteu test [40]. The absorbance was measured at 725 nm using a UV-Vis spectrophotometer (UV-1800, Shimadzu, Kyoto, Japan).

#### 2.4.4. Porosity and Density Measurements

The porosity (*Є*) and the density (*d*) of the obtained materials were determined by liquid displacement [41]. Isopropanol, as a nonsolvent of matrix-forming polymers, was liquid used in this research. The sample of the matrix was weighed (*W*) and placed in a graduated cylinder containing 3 mL of isopropanol (*V_1_*). After 5 min, the liquid level (*V_2_*) was read. The test sample was carefully removed from the cylinder and the residual isopropanol volume (*V_3_*) was recorded. This test was performed for all types of matrices in triplicate. The porosity *Є* and the density *d* of the matrices are expressed as follows:*Є (%) = (V_1_ − V_3_)/(V_2_ − V_3_) · 100*(1)
*d = W/(V_2_ − V_3_)*(2)

#### 2.4.5. Swelling Properties

A piece of each dry matrix was weighed (*W_d_*) and immersed in 5 mL phosphate buffer saline (PBS, pH = 5.7) for 15 min, 30 min, 1, 2, and 3 h. After each period, the samples were taken out from the PBS solution and weighted (*W_w_*). The test was performed in triplicate for all matrix types. The swelling ratio of matrices was defined as the ratio of increase weight to the initial weight, as follows:*swelling ratio (%) = (W_w_ − W_d_)/W_d_ · 100*(3)

#### 2.4.6. Mechanical Properties

Mechanical properties were tested using a mechanical testing machine (Z.05, Zwick/Roell, Ulm, Germany). Prior to the measurements, the cylindrical samples were measured (diameter and height). The tests were carried out at a compression speed of 50 mm/min. The Young’s modulus was calculated from the slope of the stress-strain curve in the linear region (strain from 0.05% to 0.25%). The results were recorded using the testXpert II computer program. The presented values are the average values calculated from five measurements for each type of matrices. 

#### 2.4.7. Degradation Measurements

Dry samples (*W_b_*) were weighed, placed in 12-well polystyrene plates, and immersed in 5 mL PBS (pH = 5.7). The samples were incubated at room temperature for 1, 2, 3, 7, 14, 21, and 28 days. After each period, they were removed from the PBS buffer, rinsed with deionized water three times, frozen, lyophilized, and weighed (*W_a_*). Materials were subjected to degradation measurements in triplicate. The percentage weight loss was calculated according to the following equation:*weight loss (%) = (W_b_ − W_a_)/W_b_ · 100*(4)

#### 2.4.8. In Vitro Tests

Mouse fibroblasts (3t3) were used as a model cell line to assess biomaterial in vitro cytotoxicity. Cells were cultured in a dedicated growth medium (DMEM/F12K—Corning, 10-092-CV, supplemented with 10% FBS—Corning 35-076-CV, and 1% antibiotic mixture) incubated in 5% CO_2_ and 98% relative humidity. The same incubator was also used for performing extractions and for the cytotoxicity assay. To work in accordance with the ISO 10993, biomaterial samples were weighed, cut into equal pieces not exceeding 10 mm × 5 mm in size and transferred into sterile, 50 mL sealed polypropylene centrifuge test tubes. An extraction medium identical to the medium used for cell culture of the tested cell line was used as the extraction vehicle. If the tested material exhibited characteristics of absorbent material then extra medium was added to the sample to be equal to the volume absorbed by the material. Next, a suitable amount of extraction medium was added to all tested materials (1 mL of extraction medium/0.1 g of material). The extraction was carried for 24 h in 37 °C in sterile conditions in darkness, the control and dilution medium were also incubated in the same conditions.

Cytotoxicity was assessed using a standardized MTT assay, in which the cytotoxicity of agents is evaluated based on the metabolic activity of 3t3. The MTT assay was conducted in accordance to ISO 10993 guidelines for medical devices and biomaterial testing. 3t3 cells were cultured at a density of 1 × 10^4^ cells per 100 µL of recommended growth medium on a 96 well flat bottom assay plate for 24 h prior to the addition of extracts. Before extract addition, the previous medium was discarded and all wells were rinsed with 100 µL PBS. Tested extracts were added in concentrations: 100%, 50%, 10% and 1% (v/v). Pre-incubated medium was added to the control wells (the same medium was used as extract diluting medium to ensure no bias towards the control medium). After 24 h of incubation, all wells were rinsed with 100 µL PBS, and 50 µL of MTT reagent was added at a concentration of 1 mg/mL (M5655 Sigma) for a 2 h incubation period in the CO_2_ incubator. After MTT reagent removal 100 µL of DMSO was added to all wells and the absorbance at 570 nm was measured (Multiscan Sky, Thermo, Korea). All tested compounds were compared to the correlating control wells and their values were presented as percentage of control value on charts.

## 3. Results and Discussion

### 3.1. Microspheres Morphology

Figure 1 illustrates images of the obtained microspheres. The photos were taken by the optical stereo microscope Motic SMZ-171 BLED in magnification 10×. The microspheres obtained by extrusion and emulsion methods were pictured swollen.

The presented pictures (Figure 1) show the difference between the microparticles acquired by the extrusion and the emulsion methods. The morphological observations revealed that the microspheres prepared by both methods possessed a spherical or oval shape. The swollen microparticles were characterized by a regular shape similar to the sphere and a smooth surface. 

The diameters of the obtained microparticles based on gellan gum and xanthan gum are shown in Table 2. The diameter measurements of the microspheres were made with the use of Motic SMZ-171 BLED optical microscope and scanning electron microscope. 

The largest sizes had swollen microspheres obtained by the extrusion method. Their diameters were similar and amounted to about 1200 µm, while dried microspheres, obtained by the same method, were three-times smaller (approximately 425 µm). Moreover, the microspheres sizes obtained by the emulsion method were about 220 µm and were five times smaller than microspheres prepared by extrusion. The diameters of dry microspheres obtained by the emulsion method measured using SEM were about 6 µm. Thus, we can conclude that the composition of the microspheres had a slight effect on their sizes. In contrast, the diameters of the microspheres significantly depended on the obtaining method of them.

### 3.2. Structure of Materials

The obtained SEM images of three-dimensional collagen/gelatin/hydroxyethyl cellulose composites at different magnifications are shown in Figure 2. The microspheres obtained by extrusion were not visible in the images, because they were too large and were located in the center of the matrix, which prevented their observation. The SEM images revealed that the matrices had a porous structure with irregular macropores and excellent interconnectivity. 

### 3.3. Loading Capacity of Microspheres

The loading capacity of *Calendula officinalis* flower extract into the prepared microspheres was examined using the Folin–Ciocalteu method by determining the content of polyphenolic compounds in the collected samples. 

As one can see in Figure 3, both the composition and the obtaining method of the microspheres have an impact on the effectiveness of loading extract. Comparing the composition of the prepared microspheres, we noticed that the microspheres made of gellan gum and xanthan gum had a greater ability to load *Calendula officinalis* flower extract than the gellan gum microspheres. Thus, the addition of xanthan gum to the microparticles increased the incorporation efficiency of the active substance. Moreover, the microspheres produced by emulsification showed a greater loading capacity of pot marigold extract than those produced by extrusion. The largest amount of *Calendula officinalis* flower extract was entrapped in the microspheres obtained from gellan gum and xanthan gum by emulsion method (about 39 mg/g based on gallic acid).

### 3.4. In Vitro Release

The *Calendula officinalis* flower extract release profiles from microspheres based on gellan gum (GG) and gellan gum–xanthan gum mixtures (GG +XG) in acetate buffer (pH = 5.4) at 37 °C are shown in Figure 4. 

Based on the obtained results (Figure 4), we observed that the active substance loaded in the microspheres was completely released from both types of microspheres after a maximum of 4 h. The microspheres with the addition of xanthan gum showed a slightly slower release rate of pot marigold extract than microspheres made from gellan gum, for both preparation methods. In the case of extrusion-produced microspheres, the active substance was released completely after 4 h from microspheres composed of gellan and xanthan gum mixtures, whereas from gellan gum microspheres—after 3 h. Similar observations were noted for microspheres obtained by the emulsion method. The *Calendula Officinalis* flower extract entrapped in GG + XG microspheres was released after 3 h, while the extract loaded in GG microspheres was released after 3 h. This result indicates that the addition of xanthan gum to the microspheres affects the slowdown of the release time of the substance incorporated in the microspheres. It was also noted that the obtaining method of microspheres had an impact on the release rate. Consequently, the microspheres produced by extrusion prolonged the release of the active substance.

Based on the performed analysis, it can be concluded that the modification of microparticles by the addition of xanthan gum was beneficial to control the release of the active substance. Another important observation is that the extrusion-produced microspheres release the active substance slower than the microspheres prepared by the emulsion method. This may be due to the difference in the size and surface of the microspheres. 

The explanation of our results can be that the emulsion-produced microspheres are smaller in size. Hence, the polyphenols release rate increased with decreasing microparticle dimension because of their larger surface area to volume ratio [42,43]. Chen et al. demonstrated significant differences in the release mechanism between smaller and larger microparticles, namely larger microspheres mainly determined the sustained phase of the release curve and eliminated the initial burst release [44] Belščak-Cvitanović et al. conducted research on the release of polyphenolic compounds from various plants [45]. They observed that polyphenols can easily permeate through the polymer matrix due to their relatively low molecular weight. Therefore, the release of polyphenols from gellan and xanthan gum microparticles is consistent with a diffusion-controlled release through the polymer matrix and be probably caused by bulk erosion.

### 3.5. Porosity and Density of Materials

The results of porosity and density of the prepared porous matrices are presented in Table 3. These measurements were evaluated by liquid displacement. 

All samples showed high porosity, over 83% (Table 3). The control sample had the highest porosity (*Є* = 86.6 ± 1.2%), which indicates that the addition of microspheres to the materials had a slight effect on reduction porosity due to the decrease in a number of ice crystals during the freeze-drying process of COL/GEL/HEC solutions containing microspheres [46]. It can also be seen that the matrices containing microspheres prepared by the emulsion method were characterized by higher porosity than the materials with extrusion-produced microspheres. This may be due to the fact that the microspheres obtained by the emulsion method were much smaller in size.

The control sample showed lower density (*d* = 12.7 ± 1.6 mg/mL) in comparison with materials containing the microspheres, which indicates that the addition of microspheres increases the density of the matrices (Table 3). Apart from that, there were slight differences in the density of materials between the matrices containing different amounts of microspheres. The matrices density insignificantly decreased after adding more microspheres. 

### 3.6. Swelling Tests

The swelling measurement results of collagen/gelatin/hydroxyethyl cellulose matrices are shown in Figure 5. The swelling tests were made after 15 min, 30 min, 1 h, 2 h, and 3 h of incubation in PBS buffer (pH = 5.7).

Based on the presented data, it was observed that composition, quantity and obtaining method of the microspheres had an impact on the swelling degree of the polymer matrices. The maximum swelling degree of the prepared materials with microspheres was observed after 2 h of incubation in PBS buffer, after that time it stabilized. On the other hand, the maximum swelling degree of the control sample was reached after 30 min, and it was about 3600%. The swelling ratio of all samples after 15 min was above 2300%. The matrices containing microspheres obtained by extrusion showed a higher swelling degree than matrices with incorporated microspheres prepared by the emulsion method. This could be due to the fact that the emulsification-produced microspheres were much smaller in size. Moreover, the composites with the addition of 1.5% microspheres had a higher swelling degree than matrices containing 3% of microspheres. If we compare the composition of the microspheres, we observed that the addition of xanthan gum to the gellan gum microspheres led to a decrease in the swelling degree of the materials. 

The high swelling properties are characteristic for materials with a porous structure composed of hydrophilic polymers, such as collagen, gelatin, and hydroxyethyl cellulose. Several studies showed that the materials based on gelatin, hydroxyethyl cellulose or collagen have different swelling properties ranging between 100% and 800%, up to 3700% depending on the porosity, preparation method and the composition of materials [47,48,49]. The appropriate swelling ratio is crucial in biomaterials used as wound dressings due to the good absorption of exudates. Moreover, dried materials able to absorb a large number of exudates when applied to a skin wound represents an advantage for industrial development due to their low weight and dimensions. 

### 3.7. Mechanical Properties

The results of Young’s modulus measurements are presented in Table 4. The greatest value of the compressive module had the control sample, which indicates that the addition of the microspheres into the COL/GEL/HEC matrices decreased the materials stiffness. It can also be seen that matrices with microspheres obtained by the extrusion were stiffer than matrices with microspheres prepared by the emulsion method, due to differences in microspheres diameters. Moreover, the number of added microspheres into the matrices affected their stiffness, the greater addition of microparticles resulted in greater materials stiffness. Zhang et al. found that the mechanical properties of collagen porous materials decreased with the increase in pore size [50]. The collagen matrix prepared with ice particulates having diameters of 150–250 µm showed the highest Young’s modulus, whereas the material with 425–500 µm ice particulates had the lowest Young’s modulus (20 and 11 kPa, respectively).

### 3.8. Degradation Measurements

The percentage weight loss of COL/GEL/HEC composite matrices during 28-day immersion in PBS buffer (pH = 5.7) is shown in Figure 6. 

It can be seen that the degradation of matrices with incorporated microspheres occurred rapidly within the first 7 days of samples incubation in PBS buffer, after that time the weight loss began to stabilize. Microparticles were released during the second and third days, which can be seen on the graph as a spike in the weight loss. The greatest resistance to dissolution had the control sample, because after 28 days the weight loss was about 8%, whereas the weight loss of microspheres-loaded matrices was about 40–75% depending on the type and amount of added microspheres. The materials with the addition of gellan gum and xanthan gum microspheres were degraded faster than the matrices containing gellan gum microparticles. Moreover, the greater amount of added microspheres obtained by extrusion increased the degradation rate. In contrast, the greater addition of microspheres prepared by the emulsion method led to a decrease in the degradation rate. It can be assumed that the greater weight loss in the polymer matrices containing microspheres could have been caused by their leaching out from the samples during the degradation of matrices.

### 3.9. In Vitro Tests

The percentage survivability compared to control of 3t3 cells after 24 h of exposition to biomaterial extracts acquired from MTT assay is presented in Figure 7. In accordance with ISO 10993, survivability of at least 70% compared to control has to be maintained in order to recognize a substance as non-cytotoxic. It is also a normal occurrence for cells to show increased metabolism levels after exposition to lower levels of cytotoxic agents, in this case, lower concentrations of extracts, which was observed during this experiment. 

Among the obtained materials based on collagen, gelatin and hydroxyethyl cellulose, two samples allowed for cell survivability over 70% during the MTT assay, namely the control sample (material without microparticles) and matrix with the 3% addition of gellan gum microparticles obtained by the emulsion method. The differences between the control and the 100% extracts were not statistically significant in both cases, however, a statistically significant difference was found between 1% extract and 100% extract during the testing of the matrix with the 3% addition of gellan gum microparticles prepared by the emulsion method. It might have been caused by increased proliferation rates of 3t3 cells when exposed to a low concentration of extract. 

We have observed a similar situation regarding material with 1.5% addition of microspheres based on gellan gum and xanthan gum—exposition to 1% extract also resulted in increased metabolic activity of 3t3 cells. In both, COL/GEL/HEC matrices with the 1.5% and 3% addition of microparticles based on gellan gum and xanthan gum obtained by the extrusion method, the survivability compared to control were 47.7% (±9.3) and 46.7% (±9.9), respectively. Moreover, a statistically significant difference between the control medium and the 100% extract was observed. Those results do not classify these materials as non-cytotoxic but indicate that minor changes might improve their cytotoxic properties in the future. 

## 4. Conclusions

The gellan gum and xanthan gum were used in order to obtain different types of microspheres. The spherical microparticles with loaded *Calendula officinalis* flower extract were prepared by two methods: extrusion and emulsion. The emulsion-produced microspheres were characterized by higher loading capacity than microspheres prepared by the extrusion method. The modification of gellan gum-based microspheres by adding xanthan gum into their composition had an impact on the prolonged release rate of the active substance. The prepared microspheres were incorporated into collagen/gelatin/hydroxyethyl cellulose sponges. The addition of microspheres into porous matrices resulted in a decrease in porosity and stiffness and an increase in the density of polymer materials. The obtained matrices showed a high swelling capacity. The degradation of materials occurred rapidly during the first 7 days of incubation in PBS buffer. 

Both types of prepared microspheres can be used in products with an extended-release time of the herbal extract. The modification of a polymer matrices with a porous structure by incorporation of microspheres results in potential forms of cosmetic products, which should be stored lyophilized and should be swollen with water immediately before use. COL/GEL/HEC matrix with a 3% addition of gellan gum microspheres obtained by the emulsion method was the most promising for in vitro results.

## Figures and Tables

**Figure 1 materials-13-03507-f001:**
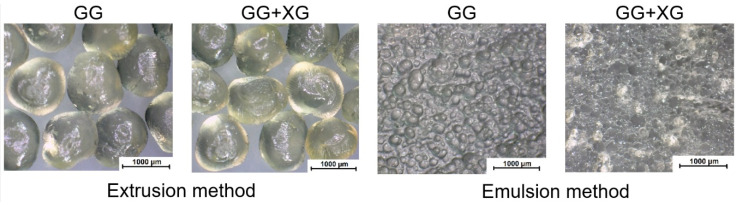
Microscope images of swollen microspheres prepared from gellan gum (GG) and gellan gum with xanthan gum (XG) by extrusion and emulsion method. Scale bar = 1000 µm.

**Figure 2 materials-13-03507-f002:**
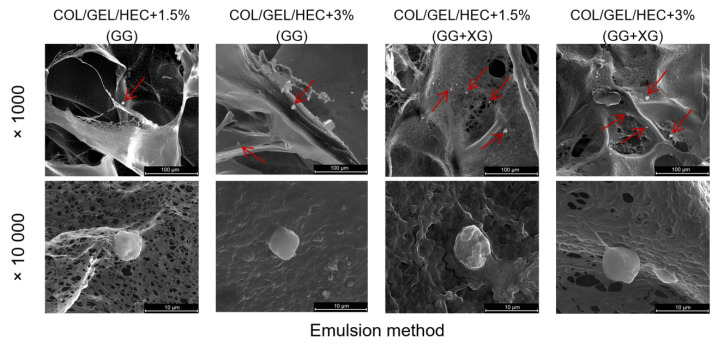
Scanning electron microscopy (SEM) images of matrices obtained by the emulsion method containing microspheres prepared from gellan gum (GG) and gellan gum with xanthan gum (GG + XG) in magnification 1000× (scale bar = 100 µm) and 10,000× (scale bar = 10 µm). The microspheres in matrices are indicated by arrows.

**Figure 3 materials-13-03507-f003:**
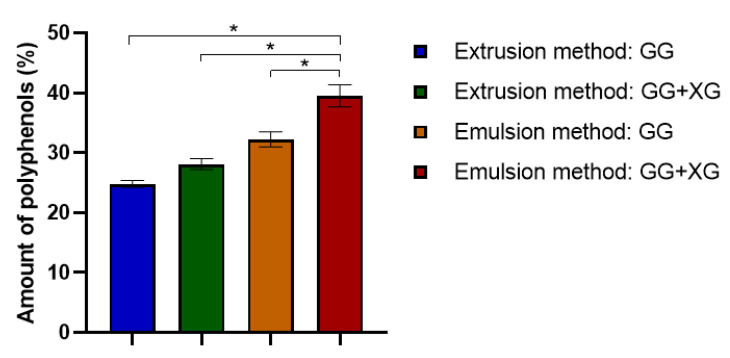
Loading capacity of microspheres based on gellan gum and gellan gum with xanthan gum mixtures. ANOVA-one way with Tukey’s post-hoc analysis (Cl = 95%) was performed to statistically compare the results. Significant differences between the results were marked on the graphs via clamps with (*).

**Figure 4 materials-13-03507-f004:**
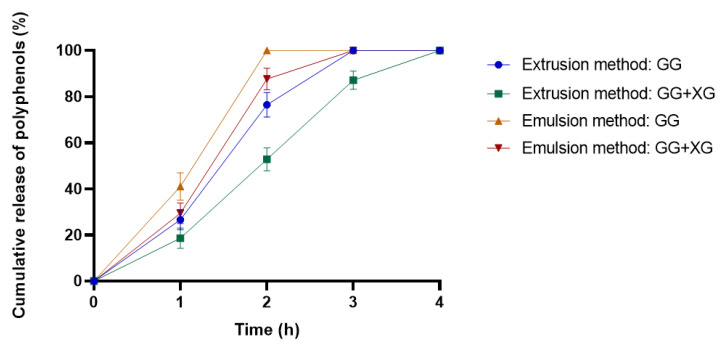
In vitro release assays of microspheres based on gellan and xanthan gum obtained by extrusion and emulsion method.

**Figure 5 materials-13-03507-f005:**
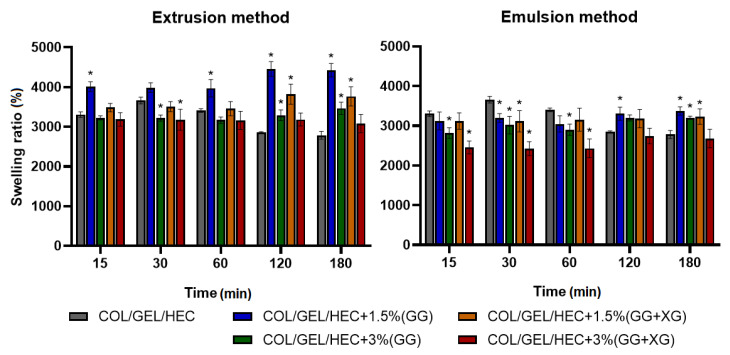
Swelling percentage of COL/GEL/HEC matrices with incorporated microspheres obtained by extrusion and emulsion method. ANOVA-one way with Dunnett’s post-hoc analysis (Cl = 95%) was performed to statistically compare the results. Significant differences compared to the control for each time were marked on the graphs with (*).

**Figure 6 materials-13-03507-f006:**
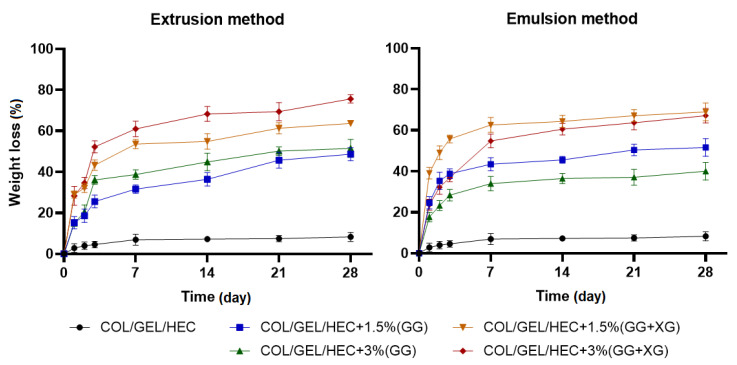
The values of weight loss during the degradation of COL/GEL/HEC matrices with incorporated microspheres obtained by extrusion and emulsion method.

**Figure 7 materials-13-03507-f007:**
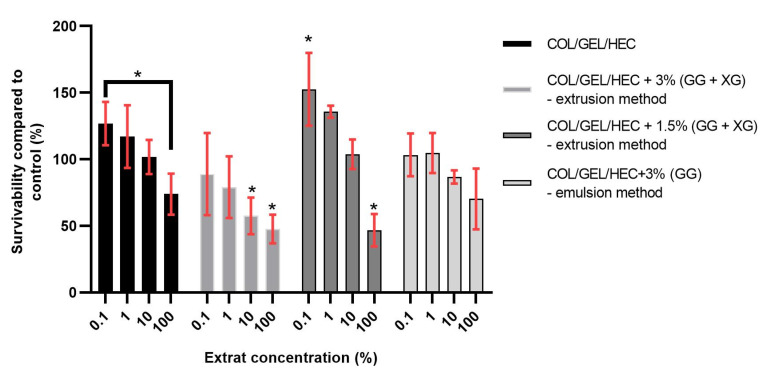
The survivability compared to the control of 3t3 cells after 24 h of exposition to COL/GEL/HEC matrices extracts acquired from MTT assay. ANOVA-one way with Dunnett’s post-hoc analysis (Cl = 95%) was performed to statistically compare the results. Significant differences compared to the control were marked on the graphs with (*), comparisons to results other than the control were indicated via clamps.

**Table 1 materials-13-03507-t001:** The types of prepared matrices composed of collagen (COL), gelatin (GEL), and hydroxyethyl cellulose (HEC) containing different amount of microspheres.

Sample	Weight Ratio (%)			Addition (%)
	COL	GEL	HEC	Microspheres
COL/GEL/HEC	25	25	50	-
COL/GEL/HEC + 1.5% (GG)	25	25	50	1.5
COL/GEL/HEC + 3% (GG)	25	25	50	3.0
COL/GEL/HEC + 1.5% (GG + XG)	25	25	50	1.5
COL/GEL/HEC + 3% (GG +XG)	25	25	50	3.0

**Table 2 materials-13-03507-t002:** Diameters of microspheres obtained by extrusion and emulsion methods.

Type of Microspheres	Size of Swollen Microspheres (µm)		Size of Dry Microspheres (µm)	
	Extrusion Method	Emulsion Method	Extrusion Method	Emulsion Method
GG	1214 ± 31	219 ± 15	384 ± 52	5.75 ± 1.16
GG + XG	1256 ± 41	226 ± 13	448 ± 23	7.05 ± 1.23

**Table 3 materials-13-03507-t003:** Porosity (*Є*) and density (*d*) of COL/GEL/HEC porous matrices containing microspheres based on gellan gum and xanthan gum.

Sample	*Є* (%)		*d* (mg/mL)	
	Extrusion Method	Emulsion Method	Extrusion Method	Emulsion Method
COL/GEL/HEC	86.6 ± 1.2		12.7 ± 1.6	
COL/GEL/HEC + 1.5% (GG)	83.3 ± 0.1	85.7 ± 0.1	16.6 ± 0.8	15.6 ± 1.9
COL/GEL/HEC + 3% (GG)	84.7 ± 2.4	85.7 ± 0.1	15.9 ± 1.4	14.7 ± 0.3
COL/GEL/HEC + 1.5% (GG + XG)	85.5 ± 1.7	85.5 ± 2.0	14.4 ± 2.7	16.4 ± 1.9
COL/GEL/HEC + 3% (GG + XG)	85.4 ± 2.0	84.1 ± 1.4	13.9 ± 1.2	15.8 ± 2.1

**Table 4 materials-13-03507-t004:** Values of compressive modulus (Emod) of COL/GEL/HEC matrices with incorporated microspheres and control sample.

Sample	*E_mod_* (kPa)	
	Extrusion Method	Emulsion Method
COL/GEL/HEC	6.97 ± 0.27	
COL/GEL/HEC + 1.5% (GG)	5.00 ± 0.69	3.41 ± 0.72
COL/GEL/HEC + 3% (GG)	5.18 ± 0.99	3.71 ± 1.22
COL/GEL/HEC + 1.5% (GG + XG)	4.71 ± 1.57	4.24 ± 1.38
COL/GEL/HEC + 3% (GG + XG)	5.01 ± 0.75	4.44 ± 1.49

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
