# Peer review of "Modification of Collagen/Gelatin/Hydroxyethyl Cellulose-Based Materials by Addition of Herbal Extract-Loaded Microspheres Made from Gellan Gum and Xanthan Gum"

_materials, 2020, doi:10.3390/ma13163507_

Round 1
Reviewer 1 Report
This manuscript reported the fabrication and characterization of polymer matrices containing microspheres for potential dermatological and cosmetic applications. In this study, detailed characterizations have been done, such as structure, swelling, mechanical properties, degradation and cytotoxicity. There are some questions to be addressed.
- Line 23, “in vitro” should be in italic.
- Why to choose gellan gum and xanthan gum as microsphere materials? What are the advantages? Please add in the Introduction part to make the manuscript more readable.
- Figure 1, please add scale bar.
- Figure 4, please do statistic analysis to indicate the significant difference.
- Line 373, the author speculated that the much greater weight loss in matrices with microspheres was caused by the microsphere leaching-out during degradation. However, compared to the weight loss (around 50%) during degradation, the addition weight of microspheres into matrices was only 1.5~3%. The explanation here seemed not rational.
- Line 403-418, the content here seems to be Conclusion rather than Discussion.
Author Response
We thank the editor and reviewers for giving us the chance to enhance our manuscript and for constructive comments, which helped us to improve this paper. Thank you for your very careful review of our paper and for the corrections and suggestions that ensued. Below, we address all comments point-by-point, discussing the subsequent modifications.
In our manuscript, words or sentences in green are changes in accordance with the Reviewers’ comments.
Reviewer 1
This manuscript reported the fabrication and characterization of polymer matrices containing microspheres for potential dermatological and cosmetic applications. In this study, detailed characterizations have been done, such as structure, swelling, mechanical properties, degradation and cytotoxicity. There are some questions to be addressed.
1. Line 23, “in vitro” should be in italic.
- Thank you, it was corrected.
2. Why to choose gellan gum and xanthan gum as microsphere materials? What are the advantages? Please add in the Introduction part to make the manuscript more readable.
- Thank you for your right comment. The section “Introduction” was supplemented by information on the gellan gum and xanthan gum.
3. Figure 1, please add scale bar.
- The scale bar was added.
4. Figure 4, please do statistic analysis to indicate the significant difference.
- In the revised manuscript the statistical analysis of obtained data was performed. Figure 4 changed the number to 3 after revising the manuscript.
5. Line 373, the author speculated that the much greater weight loss in matrices with microspheres was caused by the microsphere leaching-out during degradation. However, compared to the weight loss (around 50%) during degradation, the addition weight of microspheres into matrices was only 1.5~3%. The explanation here seemed not rational.
- Section 2.3 with the description of obtaining materials was explained in more detail.
Summarizing: Collagen solution (0.5%) and gelatin solution (0.5%) were mixed (1/1). Then, hydroxyethyl cellulose was added so that its concentration was 1% (w/v). In the next step 1.5% or 3% (w/v) of microspheres were added. In 100 ml solution we have 0.25 g of collagen, 0.25 g of gelatin, 0.5 g of hydroxyethyl cellulose and 1.5 g or 3 g microspheres. Compared to the dry mass of polymers that remains after the freeze-drying process, the proportion of microspheres in the material is quite large.
6. Line 403-418, the content here seems to be Conclusion rather than Discussion.
- Thank you. It was corrected.
Reviewer 2 Report
The presented paper is an interesting analysis of two-step release of compound from composite material. The article describes sufficient amount of experimental work and provides extensive analysis of materials.
However, there are still problems which must be improved.
- From point of view of materials science there is problem with title, which contains formulation “properties of polymer matrices”. The “polymer matrice” in materials science means only the polymer material outside of particles. However, the authors investigated properties of entire material including particles. They are not focused to the matrix.
- The title should specify, what will be described in article. There is only vague formulation “structure and properties”. It is investigated in the majority of actual scientific papers submitted to scientific journals.
- There should be discussed the adhesion of particles and matrix. How are the particles fixed in matrix? Was the adhesion somehow improved or there was a mechanical mixing of both component? The pores seem to be too large to lock the particles in the matrix?
- Please describe better the mechanism of release. It seems to me that the basic idea was that the release is two step. First step is the degradation, where the microspheres are released. Next, the active compound is released from the microspheres. There is still question. When the whole composite material will be swollen, the water will be also inside the pores. What prevents the early release of active compound, when the microspheres are still fixed in the matrix?
- The figure 7, shows only the weight loss of material during swelling degradation. There is important to describe, in which phase the microspheres are released.
Author Response
We thank the editor and reviewers for giving us the chance to enhance our manuscript and for constructive comments, which helped us to improve this paper. Thank you for your very careful review of our paper and for the corrections and suggestions that ensued. Below, we address all comments point-by-point, discussing the subsequent modifications.
In our manuscript, words or sentences in green are changes in accordance with the Reviewers’ comments.
Reviewer 2
The presented paper is an interesting analysis of two-step release of compound from composite material. The article describes sufficient amount of experimental work and provides extensive analysis of materials.
However, there are still problems which must be improved.
- From point of view of materials science there is problem with title, which contains formulation “properties of polymer matrices”. The “polymer matrice” in materials science means only the polymer material outside of particles. However, the authors investigated properties of entire material including particles. They are not focused to the matrix.
- Thank you for your thoughtful comment, the title was corrected.
- The title should specify, what will be described in article. There is only vague formulation “structure and properties”. It is investigated in the majority of actual scientific papers submitted to scientific journals.
- Thank you for comment, the title was corrected. “The title of manuscript should be concise, specific and relevant” as required by the Instructions for Authors of manuscripts for
Due to the difficulties with creating a concise title concluding all the study and analysis carried out in the manuscript, we decided to include and complete them with keywords.
Title: Modification of collagen/gelatin/hydroxyethyl cellulose-based materials by addition of herbal extract-loaded microspheres made from gellan gum and xanthan gum
Keywords: encapsulation; microspheres; Calendula officinalis flower extract; release profile; structure; stability; cytotoxicity; mechanical properties; swelling properties
- There should be discussed the adhesion of particles and matrix. How are the particles fixed in matrix? Was the adhesion somehow improved or there was a mechanical mixing of both component? The pores seem to be too large to lock the particles in the matrix?
- The microparticles were mechanically mixed with polymers solution. The base polymer used in this study to microspheres obtain is gellan gum (GG), which is an anionic polysaccharide. 1-ethyl-3-(3-dimethylaminopropyl)-carbodiimide (EDC) and N-hydroxysuccinimide (NHS) mixture is a popular crosslinking agent for collagen–based materials. EDC and NHS have the ability to mediate the amide bond formation between amino and carboxyl The gellan gum is also activated with EDC and sulfo-NHS (C. Gering at al. PLoS One. 2019; 14(8): e0221931; P. Matricardi et al. Molecules 2009, 14,3376-3391). In our study, collagen-based materials with incorporated gellan gum-based microspheres were immersed in the mixture containing, 50 mM EDC and 25 mM NHS at room temperature for 4 h. The amide bonds are formed both between the collagen functional groups and between the collagen and gellan gum functional groups.
- Please describe better the mechanism of release. It seems to me that the basic idea was that the release is two step. First step is the degradation, where the microspheres are released. Next, the active compound is released from the microspheres. There is still question. When the whole composite material will be swollen, the water will be also inside the pores. What prevents the early release of active compound, when the microspheres are still fixed in the matrix?
- The section 3.4. was rewritten: Belščak-Cvitanovićet al. conducted research on the release of polyphenols from various plants [45]. They observed that polyphenols can easily permeate through the polymer matrix due to their relatively low molecular weight. Therefore, the release of polyphenols from gellan and xanthan gum microparticles is consistent with a diffusion-controlled release through the polymer matrix and be probably caused by bulk erosion.
- The figure 7, shows only the weight loss of material during swelling degradation. There is important to describe, in which phase the microspheres are released.
- The section 3.8 was completed: Microparticles were released during the 2nd and 3rd day, which can be seen on the graph as a spike in the weight loss.
Reviewer 3 Report
The paper by Kozlowska et al examines various aspects related to the vehicular role, of a combo drug delivery system, consisting of the gellan gum or xanthan gum microparticles incorporated into porous polymer matrices. After testing the swelling capacity/degradation kinetics, the Authors suggest a bright future for the tool in drug delivery. The toxicity tests support their expectations.
Main issues:
1. What drugs can these particles carry...: with respect, Calendula officinalis flower extract does not sound like a chemical substance to me. Does the binding/swelling capacity depend on the chemical nature of the agent or this system is good for the calendula extract only?
2.... and to what cells: it is unclear why the fibroblast model is being used for the delivery system with potential implementation in dermatology. Does it not make more sense to use a keratinocyte cell line for the tests? In this case, one could think about expanding the work to look at drug adsorption by the cells.
3. The figures are not very comprehensive, as they are. Need rethinking, better labelling, stats and thorough writing of the legends.
4. I would limit the speculative blurb about "innovative and promising basis for dermatological and cosmetic purposes" and focus on realistic applications, pros and cons of the tool. I should underline that the aims declared are way too broad and non-specific. What was the exact rationale for these studies?
5. The comparisons, if any: I struggle to see if the Authors tried to compare two different technologies for producing the microparticles. If yes, please present the results in a comparative way.
Minor:
6. The Abstract does present the work well and follows the conventional structure; however, in my view it requires some re-writing to details the methods and results in a more academic way. Sentences like "Cell survival was measured with the tetrazolium salt (MTT)." need further polishing.
7. Figure 1: please add a scale bar and ensure that the legend explains the figure contents in a self-sufficient way. Ideally, a reader should be able to understand the figure contents without dipping into the main text.
8. Table 2.: add a column specifying the method.
9. Figure 2, 3: what is the message of these figures? It is always nice to have and EM image accompanying the text but what is this telling us, especially given the fact that the images represent only a fraction of experimental groups? Figure 2 does look nice aesthetically but does it represent anything new? There should be plenty EM images of the polymers around. Furthermore, please present the into given as small print in the images elsewhere in the text, if it is relevant, to ensure the readability. If the small print is not immediately relevant, it should be removed.
10. Table 3: Please explain why the linear size is much smaller than that in Table 2.
11. Figure 4: Can we talk about any differences here? Cannot see any stats.
12. Figure 5: Why does the rate of the polyphenol release go up after the first hour? This looks counter-intuitive, as the concentration goes down and so does the gradient of the chemical potential. Can the graphs representing two different manufacturing approaches be compared within a single graph and/or be presented as absolute values? I bet the extrusion method should give higher absolute figures? The data points for the figures: is the data available for emulsion method (3, 4 hours) and the extrusion method (GG, 4 hours)? It is important for comparison and, I believe, should be plotted too.
13. Figure 6: I struggle to spot much of the dynamics in the swelling data. Could you please underline the changes between the time-points somehow? Again, for the sake of comparison, it might be useful to have the data from both methods within the same graph. I realise that might be difficult with this image format but the comparison doesn't look intuitive otherwise.
14. Figure 7: there is a clear 'jump' after day 2 (four curves out of five, extrusion method). Could you please comment?
15. Figure 8: Please label the conditions straight on the figure panels. In fact, one big figure incorporating all four panels would have looked more comprehensive. From the figure, it looks that the matrix itself is more toxic than any added microparticles.
Author Response
We thank the editor and reviewers for giving us the chance to enhance our manuscript and for constructive comments, which helped us to improve this paper. Thank you for your very careful review of our paper and for the corrections and suggestions that ensued. Below, we address all comments point-by-point, discussing the subsequent modifications.
In our manuscript, words or sentences in green are changes in accordance with the Reviewers’ comments.
Reviewer 3
The paper by Kozlowska et al examines various aspects related to the vehicular role, of a combo drug delivery system, consisting of the gellan gum or xanthan gum microparticles incorporated into porous polymer matrices. After testing the swelling capacity/degradation kinetics, the Authors suggest a bright future for the tool in drug delivery. The toxicity tests support their expectations.
Main issues:
1. What drugs can these particles carry...: with respect, Calendula officinalis flower extract does not sound like a chemical substance to me. Does the binding/swelling capacity depend on the chemical nature of the agent or this system is good for the calendula extract only?
- We are extremely sorry for this. Thank you for your constructive comment. We agree with the reviewer. Our optimism regarding the suggestion of the use of the obtained materials in the transport of drugs is unjustified based on the research carried out using only the water-glycolic extract of marigold flowers. The article has been properly edited and the conclusions that were not justified by suitable research have been removed.
2.... and to what cells: it is unclear why the fibroblast model is being used for the delivery system with potential implementation in dermatology. Does it not make more sense to use a keratinocyte cell line for the tests? In this case, one could think about expanding the work to look at drug adsorption by the cells.
- Thank you for your comment on this point. In this case the material was tested in accordance to ISO 10993 (-5,-12) so that the results would be easily reproducible by other entities. There are multiple reasons to why 3t3 cells were chosen, in our experience they are the most reliable, and predictable cell line in regard to cell culture and cytotoxicity assays, they produce the most reproducible and accurate results and are part of our standard assay array. We prefer to start with them for screening and HCS purposes and always randomize the samples so that no bias towards any sample will be present. Indeed, we intend to work with other, more specific cell lines in the future that better represent the application in mid but those (like keratinocyte) often require more expensive media and are less predictable, therefore increase the time and cost significantly, lowering our output and result accuracy.
3. The figures are not very comprehensive, as they are. Need rethinking, better labelling, stats and thorough writing of the legends.
- We are extremely sorry for this. Now the figures have been appropriately revised to show more details. We believe that their quality and readability have been improved in the revised manuscript.
4. I would limit the speculative blurb about "innovative and promising basis for dermatological and cosmetic purposes" and focus on realistic applications, pros and cons of the tool. I should underline that the aims declared are way too broad and non-specific. What was the exact rationale for these studies?
- We are currently implementing a project funded by the National Science Centre. The aim goal of the research is to obtain new materials using incorporating polymer microparticles (containing active ingredients) in the three-dimensional polymer matrix with a porous structure. The project is based on the hypothesis that it is possible to obtain materials based on microspheres or microcapsules incorporated in the polymer matrix that is likely to reveal activity towards the horny layer (stratum corneum) of the skin - the outermost layer of the skin serving also as its crucial protective barrier. In the first stage, the contact of swollen polymer matrices with the skin will lead to depressing its barrier properties and, consequently, in the next stage owing to the slow degradation of microparticles, the active ingredient will be gradually released. We decided to use the Calendula officinalis flower extractas a model active compound enclosed in microparticles. We agree with the Reviewer that we have no right to write about the drug delivery system. Our materials probably can only be helpful when designing cosmetic forms, because we used safely for skin ingredients used in this industry. We wanted to potentially increase the stability and efficiency of the plant extract.
We plan to expand our research with the evaluation of the influence of the obtained materials on the skin condition with the participation of probants using instrumental analysis.
Section "Conclusion" was rewritten.
5. The comparisons, if any: I struggle to see if the Authors tried to compare two different technologies for producing the microparticles. If yes, please present the results in a comparative way.
- First, we were concentrated to compare the differences in the size and release profile of the extract from microspheres obtained by two different methods.
Second, we focused on comparing the physicochemical properties of matrices modified with the addition of these microspheres.
Minor:
6. The Abstract does present the work well and follows the conventional structure; however, in my view it requires some re-writing to details the methods and results in a more academic way. Sentences like "Cell survival was measured with the tetrazolium salt (MTT)." need further polishing.
- The abstract was rewritten.
7. Figure 1: please add a scale bar and ensure that the legend explains the figure contents in a self-sufficient way. Ideally, a reader should be able to understand the figure contents without dipping into the main text.
- Thank you for your comment. The scale bar was added.
8. Table 2.: add a column specifying the method.
- It was corrected.
9. Figure 2, 3: what is the message of these figures? It is always nice to have and EM image accompanying the text but what is this telling us, especially given the fact that the images represent only a fraction of experimental groups? Figure 2 does look nice aesthetically but does it represent anything new? There should be plenty EM images of the polymers around. Furthermore, please present the into given as small print in the images elsewhere in the text, if it is relevant, to ensure the readability. If the small print is not immediately relevant, it should be removed.
- Thank you for your constructive comment. The figures 2-3 showing the SEM photos have been completely changed. Now the results of this analysis are in Figure 2. We agree that the imaging of the control matrix did not add anything new, therefore it was removed. Moreover, the small print in the images was removed and only the scale bar was left.
10. Table 3: Please explain why the linear size is much smaller than that in Table 2.
- Diameters of swollen and dry microspheres obtained by extrusion method and the diameters of swollen microspheres produced by emulsion method were measured by the optical microscope, because they were too large, even after drying, and could not be seen on the scanning electron microscope.The diameters of dry microspheres prepared by emulsion method were measured by SEM after incorporation into polymer matrices and lyophilization process. We decided to combine the results in a common table. Now the sizes of the microspheres are summarized in Table 2.
After lyophilization, the weight loss of microparticles obtained by extrusion was ~70% and in the case of microspheres prepared by emulsion method ~95%. This is probably due to the structure of the obtained microspheres and the proportion of the polymer part and the liquid part. Microspheres obtained by the emulsion technique were characterized by higher loading capacity, which suggests that a bigger part of these particles was occupied by the liquid plant extract.
11. Figure 4: Can we talk about any differences here? Cannot see any stats.
- It is done. In the revised manuscript, statistical analysis of obtained data was performed. Figure 4 changed the number to 3 after revising the manuscript.
12. Figure 5: Why does the rate of the polyphenol release go up after the first hour? This looks counter-intuitive, as the concentration goes down and so does the gradient of the chemical potential. Can the graphs representing two different manufacturing approaches be compared within a single graph and/or be presented as absolute values? I bet the extrusion method should give higher absolute figures? The data points for the figures: is the data available for emulsion method (3, 4 hours) and the extrusion method (GG, 4 hours)? It is important for comparison and, I believe, should be plotted too.
- Thank you for your constructive comment. The explanation of our results can be that the emulsion-produced microspheres are smaller in size. Hence, polyphenols release rate increased with decreasing microparticle dimension because of their larger surface area to volume ratio.
13. Figure 6: I struggle to spot much of the dynamics in the swelling data. Could you please underline the changes between the time-points somehow? Again, for the sake of comparison, it might be useful to have the data from both methods within the same graph. I realise that might be difficult with this image format but the comparison doesn't look intuitive otherwise.
- Thank you for your comment. It was difficult to cumulate the results on one graph, the results were very illegible. That is why we have refined the graphs presented in the paper so far and completed the statistics. We believe the results are now more transparent.
14. Figure 7: there is a clear 'jump' after day 2 (four curves out of five, extrusion method). Could you please comment?
- The section 3.8 was completed: Microparticles were released during the 2nd and 3rd day, which can be seen on the graph as a spike in the weight loss.
15. Figure 8: Please label the conditions straight on the figure panels. In fact, one big figure incorporating all four panels would have looked more comprehensive. From the figure, it looks that the matrix itself is more toxic than any added microparticles.
- Thank you for comment. Comparing 100% of the extracts - it can be seen that the matrix itself has the lowest toxicity. Besides, below is a section from the table with the sample statistics and the visible lack of significant statistical differences between all 100% extracts, including the extract from the col/gel/hec matrix.
|
Tukey's multiple comparisons test |
Mean Diff, |
95,00% CI of diff, |
Significant? |
P Value |
|
COL/GEL/HEC vs. COL/GEL/HEC+3%(GG+XG) |
26,19 |
-10,84 to 63,23 |
No |
0,1927 |
|
COL/GEL/HEC vs. COL/GEL/HEC+1.5%(GG+XG) |
27,11 |
-12,48 to 66,71 |
No |
0,2126 |
|
COL/GEL/HEC vs. COL/GEL/HEC+3%(GG) |
3,62 |
-35,97 to 43,21 |
No |
0,9913 |
|
COL/GEL/HEC+3%(GG+XG) vs. COL/GEL/HEC+1.5%(GG+XG) |
0,9212 |
-36,12 to 37,96 |
No |
0,9998 |
|
COL/GEL/HEC+3%(GG+XG) vs. COL/GEL/HEC+3%(GG) |
-22,57 |
-59,61 to 14,46 |
No |
0,292 |
|
COL/GEL/HEC+1.5%(GG+XG) vs. COL/GEL/HEC+3%(GG) |
-23,49 |
-63,09 to 16,10 |
No |
0,3117 |

Round 2
Reviewer 1 Report
no
Reviewer 3 Report
The Authors have dealt with my queries adequately, the paper should be accepted now.